# Upstroke Time as a Novel Predictor of Mortality in Patients with Chronic Kidney Disease

**DOI:** 10.3390/diagnostics10060422

**Published:** 2020-06-20

**Authors:** Wen-Hsien Lee, Po-Chao Hsu, Chun-Yuan Chu, Szu-Chia Chen, Ying-Chih Chen, Meng-Kuang Lee, Hung-Hao Lee, Chee-Siong Lee, Hsueh-Wei Yen, Tsung-Hsien Lin, Wen-Chol Voon, Wen-Ter Lai, Sheng-Hsiung Sheu, Ho-Ming Su

**Affiliations:** 1Graduate Institute of Clinical Medicine, College of Medicine, Kaohsiung Medical University, Kaohsiung 807, Taiwan; cooky-kmu@yahoo.com.tw; 2Department of Internal Medicine, Kaohsiung Municipal Siaogang Hospital, Kaohsiung 812, Taiwan; scarchenone@yahoo.com.tw (S.-C.C.); 990329kmuh@gmail.com (Y.-C.C.); 980261kmuh@gmail.com (M.-K.L.); 3Faculty of Medicine, College of Medicine, Kaohsiung Medical University, Kaohsiung 807, Taiwan; pochao.hsu@gmail.com (P.-C.H.); ksmale@seed.net.tw (C.-Y.C.); lcsphk@ms18.hinet.net (C.-S.L.); hweyen@cc.kmu.edu.tw (H.-W.Y.); lth@kmu.edu.tw (T.-H.L.); wcvoon@ms2.hinet.net (W.-C.V.); wtlai@cc.kmu.edu.tw (W.-T.L.); Sheush@kmu.edu.tw (S.-H.S.); 4Division of Cardiology, Department of Internal Medicine, Kaohsiung Medical University Hospital, Kaohsiung 807, Taiwan; 960276kmuh@gmail.com

**Keywords:** upstroke time, pulse wave velocity, blood pressure, mortality, chronic kidney disease

## Abstract

Upstroke time (UT), measured from the foot-to-peak peripheral pulse wave, is a merged parameter used to assess arterial stiffness and target vascular injuries. In this study, we aimed to investigate UT for the prediction of cardiovascular and all-cause mortality in patients with chronic kidney disease (CKD). This longitudinal study enrolled 472 patients with CKD. Blood pressure, brachial pulse wave velocity (baPWV), and UT were automatically measured by a Colin VP-1000 instrument. During a median follow-up of 91 months, 73 cardiovascular and 183 all-cause mortality instances were recorded. Multivariable Cox analyses indicated that UT was significantly associated with cardiovascular mortality (hazard ratio (HR) = 1.010, *p* = 0.007) and all-cause mortality (HR = 1.009, *p* < 0.001). The addition of UT into the clinical models including traditional risk factors and baPWV further increased the value in predicting cardiovascular and all-cause mortality (both *p* < 0.001). In the Kaplan–Meier analyses, UT ≥ 180 ms could predict cardiovascular and all-cause mortality (both log-rank *p* < 0.001). Our study found that UT was a useful parameter in predicting cardiovascular and all-cause mortality in CKD patients. Additional consideration of the UT might provide an extra benefit in predicting cardiovascular and all-cause mortality beyond the traditional risk factors and baPWV.

## 1. Introduction

Cardiovascular diseases are the major causes of death in patients with chronic kidney disease (CKD) [1,2]. The causes of mortality in patients with CKD are multifactorial, including arterial stiffness, atherosclerosis, neuro-hormone activation, cardio-renal interaction, metabolic abnormality, and endothelial dysfunction [3,4,5,6,7,8,9,10,11,12]. Sympathetic nerve overactivity and increased norepinephrine levels are associated with mortality in patients with advanced CKD [3,4]. Cardio–renal interactions, including increased left ventricular mass, left atrial volume, and systemic fluid status as well as decreased left ventricular systolic function, have important impacts on mortality in advanced CKD [5,6,7,8,9,10]. Metabolic abnormalities, such as decreased adiponectin and polymorphism of vitamin D receptors, are also associated with mortality in dialysis patients [11,12]. Some biomarkers of inflammation and endothelial dysfunction, such as asymmetric dimethylarginine, are also useful predictors for mortality in dialysis populations [2,13,14,15].

Traditional atherosclerotic risk factors, including coronary artery disease, smoking, hypertension, diabetes, and congestive heart failure, have strong impacts on mortality in CKD patients [16,17,18]. Nontraditional atherosclerotic risk factors, such as increased arterial stiffness, are significantly associated with cardiovascular and all-cause mortality in patients with early to advanced CKD and sequential end-stage renal disease [19,20,21,22,23,24]. High blood pressure, progressive arterial calcification, and attenuated vascular compliance are associated with increased arterial stiffness and the presence of atherosclerotic cardiovascular disease [25]. Vascular endothelial growth factors, norepinephrine, and proinflammatory cytokines are also significantly associated with atherogenesis in patients with CKD and end-stage renal disease [2,13,26,27,28]. Progressive arterial stiffness increases, in terms of increased brachial-ankle pulse wave velocity (baPWV), are also associated with increased mortality in patients with CKD [29].

The upstroke time (UT) is measured from the initial notch to the peak of the peripheral arterial pulse wave. In recent studies, UT was not only used as a diagnostic tool for peripheral artery disease and target organ damage but also as a predictor of cardiovascular and all-cause mortality [30,31,32]. However, the relationship of UT with mortality in CKD patients is still unknown. Therefore, we aim to explore the survival impact of UT among CKD patients.

## 2. Materials and Methods

### 2.1. Study Subjects and Design

All study participants were enrolled from patients booked for echocardiographic examinations at Kaohsiung Municipal Siaogang Hospital from March 2010 to March 2012 due to suspected ischemic heart disease, heart failure, hypertension, abnormal cardiac physical examination, a survey for dyspnea, a pre-operative cardiac function survey, and so on. Patients classified in stages 3, 4, and 5 CKD (based on the estimated glomerular filtration rate (eGFR) level (30 to 59, 15 to 29, and <15 mL/min/1.73m^2^) with kidney damage lasting for more than 3 months) were included [33]. As the value of UT had a beat-to-beat variation in patients with atrial fibrillation (*n* = 6) and severe aortic stenosis (*n* = 2) might have a significant impact on the value of UT, we excluded such patients [34,35]. In addition, we also excluded patients with end-stage renal disease receiving renal replacement therapies (*n* = 4). In total, 472 patients joined this study (Figure 1).

### 2.2. Ethics Statement

The study protocol was approved by the Institutional Review Board of Kaohsiung Medical University Hospital, and all enrolled patients gave written, informed consent to participate in the study (KMUHIRB-E(I)-20190415, 6 March 2020).

### 2.3. Blood Pressure, baPWV, and UT Measurements

Blood pressures, baPWV, and UT were measured by an instrument (Colin VP1000, Komaki, Japan) in all participants in the supine position after at least 10 min rest [18,36,37]. Detailed measurements of blood pressures and baPWV were published in our previous studies [18,38]. Briefly, the baPWV was counted as the traveled distance of the pulse wave divided by the passage time of the pulse wave from the brachial to tibial arteries. The UT was also automatically calculated as the time interval from foot-to-peak of the ankle arterial pulse wave in each participant (Figure 2). After obtaining the bilateral values, the higher values of blood pressures, baPWV, and UT were used for further analyses [30].

### 2.4. Collection of Clinical Characteristics, Medical, and Laboratory Data

The clinical characteristics and medical profiles, including the age, gender, history of smoking, hypertension, diabetes mellitus, congestive heart failure, coronary artery disease, and cerebrovascular disease were obtained from the medical records. The body mass index was calculated as a standard ratio of weight (kilograms) divided by the square of height (meters). The medications taken during this study period, including angiotensin-converting enzyme inhibitors (ACEIs), angiotensin II receptor blockers (ARBs), β-blockers, calcium channel blockers (CCBs), and diuretics, were also reviewed from the medical records. Standard laboratory data for fasting blood samples and serum creatinine were measured by an automatic analyzer (Roche Diagnostics, Mannheim, Germany). The value of eGFR was calculated using the four variable equations from the Modification of Diet in Renal Disease study [33]. 

### 2.5. Definition of Cardiovascular and All-Cause Mortality

All study participants were followed-up until December 2018. Survival information and causes of death were obtained from the official death certificate and final confirmation by the Ministry of Health and Welfare. The causes of death were classified by the International Classification of Diseases 9th Revision. The causes of cardiovascular mortality were defined as deaths due to cerebral vascular disease, ischemic heart disease, myocardial infarction, heart failure, valvular heart disease, and atherosclerotic vascular disease.

### 2.6. Statistical Analysis

All variables were presented as percentages or mean ± standard deviation. After the normality of continuous variables was determined using the Kolmogorov–Smirnov test, appropriate parametric and nonparametric tests were used. Differences between two categorical variables were analyzed by the Chi-square test. The time to mortality events were modeled using the Cox proportional hazards model with forward selection. Before multivariable analysis, we performed a collinearity test. Due to the high collinearity between systolic blood pressure and pulse pressure, only pulse pressure was selected for the multivariable analysis. We conducted two multivariable models. The variables in model 1 included the significant clinical variables in the univariable analysis except for systolic blood pressure and baPWV, and the variables in model 2 included the significant clinical variables in the univariable analysis except for systolic blood pressure and UT. The incremental values of baPWV and UT were studied over a clinical model to assess the risk for mortality events by calculating the improvement in the global Chi-square. The best cut-off value of UT for the prediction of cardiovascular and all-cause mortality was determined by a receiver operating characteristic curve. A Kaplan–Meier survival plot was calculated from the baseline to time of mortality events and compared using the log-rank test. The statistical significance was defined as a *p* value less than 0.05. Statistical analysis was performed using SPSS version 22.0 (SPSS, Chicago, IL, USA).

## 3. Results

### 3.1. Baseline Characteristics in All Participants

The difference of clinical characteristics between patients with and without all-cause mortality is shown in all-cause mortality in Table 1. Patients with all-cause mortality had an older age, a higher prevalence of diabetes, cerebrovascular disease, and congestive heart failure, a higher pulse pressure, heart rate, fasting glucose, baPWV, and UT, a higher percentage of using diuretics and advanced CKD (eGFR < 30 mL/min/1.73m^2^), and a lower body mass index, total cholesterol, and eGFR. 

### 3.2. Major Predictors of All-Cause and Cardiovascular Mortality in Study Patients 

During the follow-up period (median 91 months), 73 cardiovascular mortality and 183 all-cause mortality instances were recorded. Table 2 shows the predictors of cardiovascular and all-cause mortality in univariate analysis. Increased cardiovascular mortality was significantly associated with increased age, heart rate, serum fasting glucose, baPWV, and UT (hazard ratio (HR) = 1.015; 95% confidence interval (CI) = 1.010 to 1.020; *p* < 0.001), decreased body mass index and eGFR; and the presence of diabetes, coronary artery disease, cerebrovascular disease, and congestive heart failure. Increased all-cause mortality was significantly associated with increased age, systolic blood pressure, heart rate, serum fasting glucose, baPWV, and UT (HR = 1.014, 95% CI= 1.011 to 1.017; *p* < 0.001), decreased body mass index, total cholesterol, and eGFR, the presence of diabetes, cerebrovascular disease, and congestive heart failure, and the use of diuretics.

In Table 3 and Table 4, we performed a collinearity test. Both in cardiovascular and all-cause mortality analysis, the values of the variance inflation factors of systolic blood pressure and pulse pressure were greater than four (Table 3). After we withdrew the variable of systolic blood pressure, all values of the variance inflation factors in cardiovascular and all-cause mortality analysis were less than four (Table 4). Due to the high collinearity between systolic blood pressure and pulse pressure, only pulse pressure was selected into the multivariable models.

Table 5 presents the multivariable Cox forward hazards analyses of cardiovascular and all-cause mortality. In model 1, increased cardiovascular mortality was significantly associated with increased age, the presence of coronary artery disease and congestive heart failure, and increased baPWV (HR = 1.011 per 10 cm/s, 95% CI = 1.003 to 1.019, *p* = 0.008). Meanwhile, increased all-cause mortality was significantly associated with increased age, decreased eGFR, the presence of cerebrovascular disease and congestive heart failure, and baPWV (HR = 1.010 per 10 cm/s, 95% CI = 1.005 to 1.015, *p* < 0.001). In model 2, increased cardiovascular mortality was significantly associated with increased age, the presence of congestive heart failure, and increased UT (HR = 1.010, 95% CI = 1.003 to 1.017, *p* = 0.007). Meanwhile, increased all-cause mortality was significantly associated with increased age, decreased eGFR, the presence of cerebrovascular disease and congestive heart failure, and increased UT (HR = 1.009, 95% CI = 1.004 to 1.013, *p* < 0.001).

From the receiver operating characteristic curve (Figure 3), the best cut-off value of UT was 180 ms with 46.6% sensitivity and 80.7% specificity for the prediction of cardiovascular mortality and 43.2% sensitivity and 88.9% specificity for the prediction of all-cause mortality. Figure 4 shows the Kaplan–Meier curves for cardiovascular-mortality-free and all-cause-mortality-free survival between patients with UT ≥ 180 ms versus UT < 180 ms (both log-rank *p* < 0.001).

The incremental value of UT in mortality prediction is shown in Figure 5. The variables in the clinical models included the variables that were significant in the multivariable analyses of both model 1 and model 2, except the baPWV and UT. Hence, the variables in the clinical model for cardiovascular mortality prediction included age and the presence of congestive heart failure, and the variables in the clinical model for all-cause mortality prediction included age, eGFR, and the presence of cerebrovascular disease and congestive heart failure. The clinical models could significantly predict the cardiovascular (Chi-square = 71.145, *p* < 0.001) and all-cause mortality (Chi-square = 141.683, *p* < 0.001). The addition of UT into the clinical models plus baPWV offered an extra benefit in the prediction of cardiovascular (Chi-square increase 23.155, *p* < 0.001) and all-cause mortality (Chi-square increase 28.439, *p* < 0.001). 

The variables in the clinical model for cardiovascular mortality prediction included age and the presence of congestive heart failure. The variables in the clinical model for all-cause mortality prediction included age, estimated glomerular filtration rate, and the presence of cerebrovascular disease and congestive heart failure. In the incremental models, the addition of upstroke time into the clinical models plus brachial-ankle pulse wave velocity (baPWV) resulted in a significant improvement in the predictive value for cardiovascular mortality (A) and all-cause mortality (B) in patients with chronic kidney disease (both *p* < 0.001). 

## 4. Discussion

There were two major findings in the present study. First, increased UT was associated with increased cardiovascular and all-cause mortality in CKD patients. Second, additional consideration of UT might provide an extra benefit in predicting cardiovascular and all-cause mortality beyond the traditional risk factors and baPWV. 

Our present study is the first investigation to use UT as a useful parameter in predicting cardiovascular and all-cause mortality in CKD patients. Although the major role of UT on mortality has not been well-understood, some mechanisms might be able to explain the association between the prolongation of UT and poor prognosis. First, UT could be served a surrogate maker for atherosclerotic vascular disease and target organ damage [31,32]. By physiological definition, UT was the time interval from the foot to the peak of the arterial wave. Increased UT was associated with atherosclerotic vascular disease, such as peripheral vascular disease and coronary artery disease [31,32]. Atherosclerosis, the presence of carotid plaques, and coronary artery calcification were significantly correlated with a high mortality rate in CKD patients [39,40,41]. Consequently, the association between the prolongation of UT and atherosclerosis might explain increased UT as a poor prognostic parameter in CKD patients in the present study. UT, UT per cardiac cycle, and the combination of UT and the ankle-brachial index had a significant influence on the diagnosis of early peripheral artery disease [31,32,42]. Recently, Shoji et al. demonstrated that UT was higher in patients with coronary artery disease than in subjects without coronary artery disease. Increased UT was significantly associated with the severity of coronary calcification in elder populations with coronary artery disease [43]. Therefore, the impact of UT on mortality might be through the atherosclerotic processes in CKD populations.

In addition, UT could be a parameter of left ventricular function and geometry [30,44,45]. Abnormal cardiac function, including left ventricular hypertrophy, reduced left ventricular ejection fraction, and increased left ventricular filling pressure and wall stress, have been associated with poor cardiovascular outcomes in patients with CKD [46,47,48]. Previous studies have shown that increased UT was associated with decreased left ventricular ejection fraction, elevated left ventricular end-diastolic pressure, increased left ventricular systolic wall stress, and left ventricular hypertrophy [30,44,45]. Therefore, the association of increased UT with poor cardiac function and abnormal left ventricular geometry might partially explain why the prolongation of UT and UT ≥ 180 ms were useful in the prediction of increased cardiovascular and all-cause mortality in CKD patients. From previous studies, baPWV was not only a marker for arterial stiffness and left ventricular diastolic function but also a predictor for all-cause mortality [29,36,49,50,51,52]. Possible mechanisms of baPWV-associated mortality could be through vascular dysfunction, left ventricular hypertrophy, and left ventricular diastolic dysfunction [20,52]. In our present study, adding UT into a basic clinical model including traditional risk factors plus baPWV could provide an incremental value for the prediction of cardiovascular and all-cause mortality. Hence, the additional consideration of UT might be useful in predicting cardiovascular and all-cause mortality over traditional risk factors and baPWV in patients with CKD. 

## 5. Study Limitations

Our present study had some limitations. First, as the study subjects were already being evaluated for heart disease by echocardiography, the study was susceptible to selection bias, making the findings potentially less-generalized; Second, owing to the study design and ethical concerns, we did not withdraw hypertension medications. Our study result might be affected by these medications; Third, patients with acute renal damage, eGFR > 60 mL/min/1.73m^2^, and end-stage renal disease under hemodialysis or peritoneal dialysis therapies were excluded. Therefore, our study findings cannot be applied to such patients; Fourth, because patients who were excluded did not receive peripheral vascular examination, we could not provide the results of all variable comparisons between those who were excluded and those who were included; Finally, several biomarkers including cardiac natriuretic peptide, C reactive protein, and asymmetric dimethylarginine were lacking in our study, so our present study was limited in providing pathophysiologic evidence of mortality [14,23,28].

## 6. Conclusions

In this study, we found UT to be a useful parameter for predicting cardiovascular and all-cause mortality in CKD. Additional consideration of UT might provide an extra benefit in predicting cardiovascular and all-cause mortality beyond the traditional risk factors and baPWV.

## Figures and Tables

**Figure 1 diagnostics-10-00422-f001:**
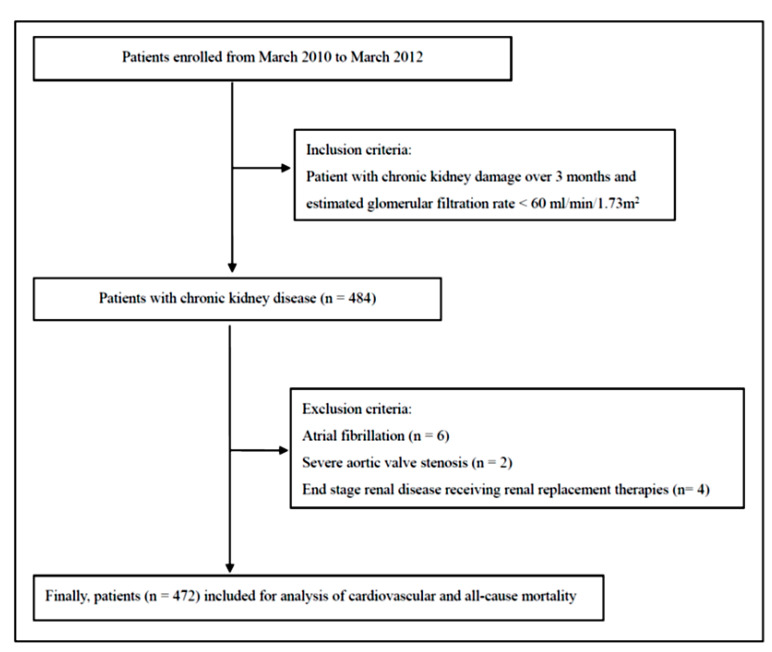
Flowchart of the study protocol.

**Figure 2 diagnostics-10-00422-f002:**
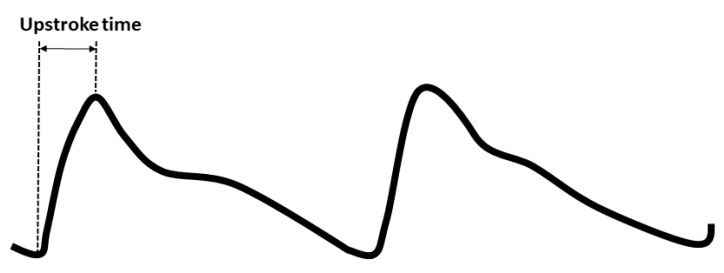
The upstroke time was automatically calculated from the foot-to-peak of peripheral pulse wave.

**Figure 3 diagnostics-10-00422-f003:**
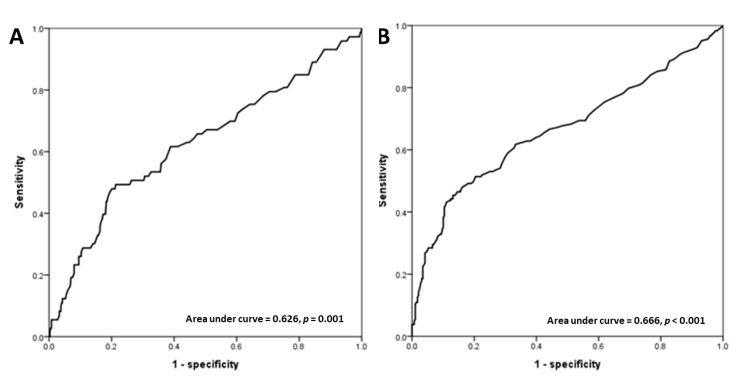
The receiver operating characteristic curves for cardiovascular mortality (**A**) and all-cause mortality (**B**) prediction in our study patients.

**Figure 4 diagnostics-10-00422-f004:**
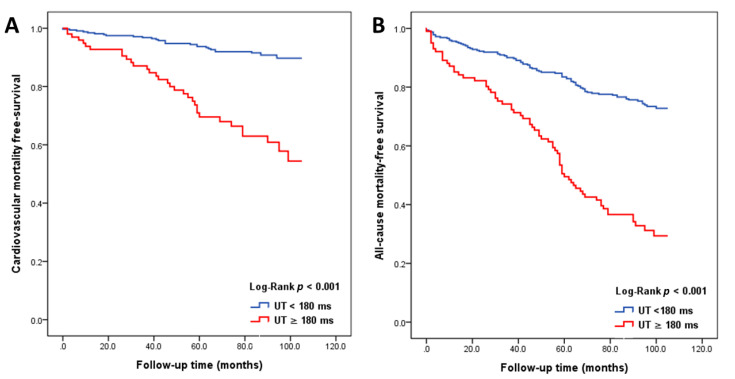
Kaplan–Meier analyses for cardiovascular-mortality-free survival (**A**) and all-cause-mortality-free survival (**B**) in patients with chronic kidney disease.

**Figure 5 diagnostics-10-00422-f005:**
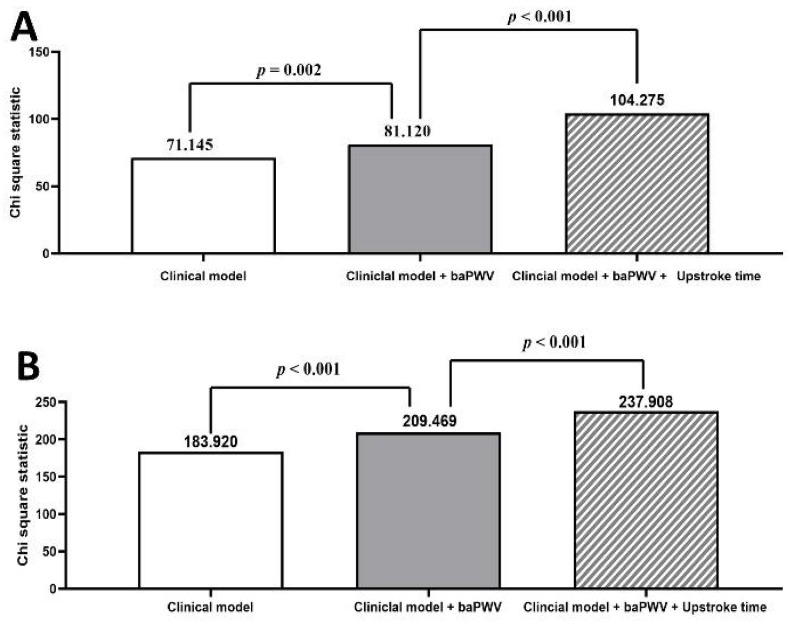
Incremental models for cardiovascular mortality (**A**) and all-cause mortality (**B**) in patients with chronic kidney disease.

**Table 1 diagnostics-10-00422-t001:** Comparison of the clinical characteristics between patients with and without all-cause mortality.

Variables	Patients with All-Cause Mortality (*n* = 183)	Patients Without All-Cause Mortality (*n* = 289)	*p* Value *	All Patients (*n* = 472)	*p* Value **
Age (years)	72.4 ± 11.4	63.3 ± 11.6	<0.001	66.8 ± 12.4	<0.001
Male gender (%)	54.6	49.1	0.243	51.3	
Smoking history (%)	8.2	11.3	0.303	10.1	
Diabetes mellitus (%)	49.2	28.7	<0.001	36.7	
Hypertension (%)	77.0	75.1	0.627	75.8	
Coronary artery disease (%)	23.6	17.0	0.078	19.6	
Cerebrovascular disease (%)	15.1	4.0	<0.001	8.1	
Congestive heart failure (%)	25.2	4.4	<0.001	12.0	
Systolic blood pressure (mmHg)	146 ± 25	140 ± 20	0.013	142 ± 22	0.001
Pulse pressure (mmHg)	69 ± 18	61 ± 13	<0.001	78 ± 13	<0.001
Heart rate (beats/min)	73 ± 15	69 ± 12	0.003	70 ± 13	<0.001
Body mass index (kg/m^2^)	25.4 ± 4.1	26.6 ± 4.0	0.001	26.1 ± 4.1	0.005
Antihypertensive medication					
ACEI (%)	12.6	8.0	0.100	9.7	
ARB (%)	47.5	55.0	0.113	52.1	
β blocker (%)	42.1	45.5	0.468	44.2	
Calcium channel blocker (%)	43.7	48.8	0.282	46.8	
Diuretics (%)	46.7	31.0	0.001	37.1	
Laboratory parameters					
Fasting glucose (mg/dL)	125.4 ± 51.0	112.4 ± 40.3	0.007	116.9 ± 44.7	<0.001
Triglyceride (mg/dL)	151.5 ± 101.1	153.0 ± 102.6	0.693	152.5 ± 102.0	<0.001
Total cholesterol (mg/dL)	183.0 ± 40.7	194.1 ± 41.2	0.008	190.1 ± 41.3	0.002
Base line eGFR (mL/min/1.73 m^2^)	35.3 ± 15.9	45.9 ± 13.4	<0.001	41.8 ± 15.3	<0.001
eGFR< 30 mL/min/1.73 m2 (%)	36.1	12.1	<0.001	21.4	
baPWV (cm/s)	2128.1 ± 599.4	1776.1 ± 356.5	<0.001	1912.6 ± 496.0	<0.001
Upstroke time (ms)	177.4 ± 47.6	150.3 ± 28.0	<0.001	160.8 ± 39.1	<0.001

Abbreviations. ACEI: angiotensin converting enzyme inhibitor; ARB: angiotensin II receptor blocker; baPWV: brachial-ankle pulse wave velocity; eGFR: estimated glomerular filtration rate. *p* value * for two group comparison; *p* value ** for Kolmogorov–Smirnov test.

**Table 2 diagnostics-10-00422-t002:** Predictors of cardiovascular and all-cause mortality in univariate analysis among all study patients.

Variables	Cardiovascular Mortality		All-Cause Mortality	
	HR (95% CI)	*p* Value	HR (95% CI)	*p* Value
Age (years)	1.067 (1.040, 1.095)	<0.001	1.072 (1.054, 1.090)	<0.001
Gender (male)	0.863 (0.523, 1.423)	0.564	1.043 (0.760, 1.433)	0.794
Smoking history	1.076 (0.463, 2.497)	0.865	0.806 (0.456, 1.422)	0.456
Diabetes mellitus	2.292 (1.391, 3.778)	0.001	1.955 (1.422, 2.687)	<0.001
Hypertension	0.707 (0.415, 1.205)	0.202	1.019 (0.706, 1.471)	0.919
Coronary artery disease	2.184 (1.282, 3.721)	0.004	1.307 (0.893, 1.914)	0.169
Cerebrovascular disease	3.193 (1.620, 6.297)	0.001	2.757 (1.753, 4.336)	<0.001
Congestive heart failure	5.050 (2.908, 8.771)	<0.001	3.635 (2.487, 5.313)	<0.001
Systolic blood pressure (per 1 mmHg)	1.007 (0.996, 1.019)	0.207	1.010 (1.002, 1.017)	<0.001
Pulse pressure (per 1 mmHg)	1.024 (1.008, 1.040)	0.003	1.027 (1.017, 1.037)	<0.001
Heart rate (per 1 beats/min)	1.023 (1.004, 1.041)	0.015	1.017 (1.005, 1.029)	0.005
Body mass index (per 1 kg/m^2^)	0.912 (0.849, 0.979)	0.011	0.934 (0.893, 0.976)	0.002
Antihypertensive medication				
ACEI	0.753 (0.273, 2.074)	0.583	1.195 (0.701, 2.036)	0.513
ARB	1.096 (0.663, 1.810)	0.721	0.835 (0.608, 1.146)	0.264
β blocker	1.158 (0.703, 1.905)	0.565	0.962 (0.699, 1.324)	0.811
Calcium channel blocker	0.957 (0.581, 1.576)	0.862	0.917 (0.667, 1.260)	0.593
Diuretics	1.274 (0.766, 2.117)	0.350	1.670 (1.216, 2.295)	0.002
Laboratory parameters				
Fasting glucose (per 1 mg/dL)	1.007 (1.002, 1.012)	0.007	1.005 (1.001, 1.008)	0.008
Triglyceride (per 1 mg/dL)	0.999 (0.996, 1.002)	0.502	0.999 (0.998, 1.001)	0.597
Total cholesterol (per 1 mg/dL)	0.994 (0.987, 1.002)	0.127	0.992 (0.998, 0.997)	0.001
Baseline eGFR (per 1 mL/min/1.73 m^2^)	0.979 (0.964, 0.994)	0.006	0.967 (0.958, 0.976)	<0.001
baPWV (per 10 cm/s)	1.011 (1.007, 1.016)	<0.001	1.013 (1.010, 1.016)	<0.001
Upstroke time (per 1 ms)	1.015 (1.010, 1.020)	<0.001	1.014 (1.011, 1.017)	<0.001

Abbreviations. ACEI: angiotensin converting enzyme inhibitor; ARB: angiotensin II receptor blocker; baPWV: brachial-ankle pulse wave velocity; CI: confidence interval; eGFR: estimated glomerular filtration rate; HR: hazard ratio.

**Table 3 diagnostics-10-00422-t003:** Collinear analysis for all continuous variables in cardiovascular and all-cause mortality models.

Continuous Variables	Variance Inflation Factors in Cardiovascular Mortality Model	Variance Inflation Factors in All-Cause Mortality Model
Age (years)	1.925	1.925
Systolic blood pressure (mmHg)	4.927	4.927
Pulse pressure (mmHg)	5.388	5.388
Heart rate (beats/min)	1.349	1.349
Body mass index (kg/m^2^)	1.099	1.099
Fasting glucose (mg/dL)	1.098	1.098
Triglyceride (mg/dL)	1.115	1.115
Total cholesterol (mg/dL)	1.128	1.128
Baseline eGFR (mL/min/1.73 m^2^)	1.127	1.127
Brachial-ankle pulse wave velocity (per 10 cm/s)	2.010	2.010
Upstroke time (ms)	1.546	1.546

**Table 4 diagnostics-10-00422-t004:** Collinear analysis for all continuous variables after removing systolic blood pressure in cardiovascular and all-cause mortality models.

Continuous Variables (After Removing Systolic Blood Pressure)	Variance Inflation Factors in Cardiovascular Mortality Model	Variance Inflation Factors in All-Cause Mortality Model
Age (years)	1.560	1.560
Pulse pressure (mmHg)	1.734	1.734
Heart rate (beats/min)	1.312	1.312
Body mass index (kg/m^2^)	1.094	1.094
Fasting glucose (mg/dL)	1.090	1.090
Triglyceride (mg/dL)	1.115	1.115
Total cholesterol (mg/dL)	1.127	1.127
Baseline eGFR (mL/min/1.73 m^2^)	1.127	1.127
Brachial-ankle pulse wave velocity (per 10 cm/s)	1.727	1.727
Upstroke time (ms)	1.469	1.469

**Table 5 diagnostics-10-00422-t005:** Predictors of cardiovascular and all-cause mortality in multivariate analysis among all study patients.

Variables	Cardiovascular Mortality				All-Cause Mortality			
	Model 1 HR (95% CI)	*p* Value	Model 2 HR (95% CI)	*p* Value	Model 1 HR (95% CI)	*p* Value	Model 2 HR (95% CI)	*p* Value
Age (per 1 year)	1.081 (1.039, 1.124)	<0.001	1.007 (1.035, 1.020)	<0.001	1.069 (1.043, 1.094)	<0.001	1.070 (1.045, 1.096)	<0.001
Coronary artery disease	2.433 (1.257, 4.708)	0.008	1.523 (0.755, 3.073)	0.240	/		/	
Cerebrovascular disease	1.863 (0.747, 4.647)	0.182	1.782 (0.709, 4.481)	0.219	2.251 (1.232, 4.113)	0.008	2.235 (1.232, 4.054)	0.008
Congestive heart failure	6.776 (3.276, 14.015)	<0.001	5.199 (2.546, 10.617)	<0.001	3.056 (1.817, 5.138)	<0.001	2.291 (1.379, 3.808)	0.001
Fasting glucose (per 1 mg/dL)	/		/		1.003 (0.999, 1.007)	0.151	1.003 (0.944, 0.971)	0.090
Baseline eGFR (per 1 mL/min/1.73 m^2^)	/		/		0.963 (0.950, 0.977)	<0.001	0.957 (0.944, 0.971)	<0.001
baPWV (per 10 cm/s)	1.011 (1.003, 1.019)	0.008	/		1.010 (1.005, 1.015)	<0.001	/	
Upstroke time (per 1 ms)	/		1.010 (1.003, 1.017)	0.007	/		1.009 (1.004, 1.013)	<0.001

Abbreviations. baPWV: brachial-ankle pulse wave velocity; CI: confidence interval; eGFR: estimated glomerular filtration rate; HR: hazard ratio.

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
