# Peer review of "Upstroke Time as a Novel Predictor of Mortality in Patients with Chronic Kidney Disease"

_diagnostics, 2020, doi:10.3390/diagnostics10060422_

Round 1

Reviewer 1 Report

This paper by Lee et al. entitled “Upstroke time as a novel predictor of mortality in patients with chronic kidney disease” focused on the relationship between foot-to-peak peripheral pulse wave and mortality in patients with 3-5 KDOKI stages of CKD. Cardiovascular risk (CV) in CKD is known to be much higher than that of the reference population, particularly in End Stage Renal Disease (ESRD). This topic has received great attention in the last 20 years, but in the reference list of this manuscript several papers which have contributed to make the scientific history in this field are unfortunately missing. The main determinants of cardiovascular risk, particularly in more advanced stages of CKD, were found to be volume overload associated with the development of cardiac remodeling/dysfunction (Zoccali C. et al., J Am Soc Nephrol. 2001 Jul;12(7):1508-15; Mallamaci F. et al. Kidney Int. 2001 Apr;59(4):1559-66; Cataliotti A. et al. Mayo Clin Proc. 2001 Nov;76(11):1111-9; Enia G. et al. Nephrol Dial Transplant 2001 Jul;16(7):1459-64; Zoccali C. et al. J Am Soc Nephrol. 2001 Dec;12(12):2768-74; Zoccali C. et al. Kidney Int. 2002 Jul;62(1):339-45; Zoccali C. et al. Kidney Int. 2004 Apr;65(4):1492-8; Zoccali C. et al. J Am Soc Nephrol. 2004 Apr;15(4):1029-37; Zoccali C. et al. J Am Soc Nephrol. 2006 May;17(5):1460-5; Tripepi G. et al. J Am Soc Nephrol. 2007 Apr;18(4):1316-22; Mallamaci F. et al. J Hypertens 2008 Sep;26(9):1875-82; Tripepi G. et al. Hypertension 2009 Oct;54(4):818-24; Abd ElHafeez S. et al. J Nephrol. 2015 Dec;28(6):729-37 ), corroborated by sympathetic activation (Zoccali C. et al. Circulation. 2002 Mar 19;105(11):1354-9), inflammation (Zoccali C. et al. J Am Soc Nephrol. 2002 Feb;13(2):490-6; Mallamaci F. et al. Kidney Int. 2005 Jun;67(6):2330-7.Tripepi G. et al. Clin J Am Soc Nephrol. 2011 Jul;6(7):1714-21;), as well as a metabolic derangement (Zoccali C. et al. Lancet 2001; 358(9299):2113-7; Zoccali C. et al., J Am Soc Nephrol 2002 Jan;13(1):134-41; Mallamaci F. et al. J Am Soc Nephrol. 2004 Feb;15(2):435-41; Testa A. et al. J Bone Miner Res. 2010 Feb;25(2):313-9. These determinants of CV in CKD are no doubt involved in the pathophysiology of arterial stiffness, that is the topic of this paper. Therefore, previous bibliography cannot be omitted in the reference list and should be brought to the fore both in the Introduction and in the Discussion. As to upstroke time, that is a novel predictor of mortality in patients with chronic kidney disease, it looks promising, although whether it provides additional information as compared with pulse wave velocity (PWV) remains so far unclear. Specific Comments 1. Does upstroke time have an additional value as compared with PWV? 2. PWV was shown to be predicted by pulse pressure in patients with mild to moderate CKD (Stancanelli B. et al. Kidney Blood Press Res. 2007;30(5):283-8). What about the relationship between pulse pressure and upstroke time in your patients with CKD? 3. Did you find any difference in upstroke time as related to different classes of drugs (inhibitors of Renin-Angiotensin Aldosterone System, Calcium Antagonists and Beta-Blockers)?

Author Response

The comments of the reviewer 1 are as follows:

Review 1 comment

 This paper by Lee et al. entitled “Upstroke time as a novel predictor of mortality in patients with chronic kidney disease” focused on the relationship between foot-to-peak peripheral pulse wave and mortality in patients with 3-5 KDOKI stages of CKD. Cardiovascular risk (CV) in CKD is known to be much higher than that of the reference population, particularly in End Stage Renal Disease (ESRD). This topic has received great attention in the last 20 years, but in the reference list of this manuscript several papers which have contributed to make the scientific history in this field are unfortunately missing. The main determinants of cardiovascular risk, particularly in more advanced stages of CKD, were found to be volume overload associated with the development of cardiac remodeling/dysfunction (Zoccali C. et al., J Am Soc Nephrol. 2001 Jul;12(7):1508-15; Mallamaci F. et al. Kidney Int. 2001 Apr;59(4):1559-66; Cataliotti A. et al. Mayo Clin Proc. 2001 Nov;76(11):1111-9; Enia G. et al. Nephrol Dial Transplant 2001 Jul;16(7):1459-64; Zoccali C. et al. J Am Soc Nephrol. 2001 Dec;12(12):2768-74; Zoccali C. et al. Kidney Int. 2002 Jul;62(1):339-45; Zoccali C. et al. Kidney Int. 2004 Apr;65(4):1492-8; Zoccali C. et al. J Am Soc Nephrol. 2004 Apr;15(4):1029-37; Zoccali C. et al. J Am Soc Nephrol. 2006 May;17(5):1460-5; Tripepi G. et al. J Am Soc Nephrol. 2007 Apr;18(4):1316-22; Mallamaci F. et al. J Hypertens 2008 Sep;26(9):1875-82; Tripepi G. et al. Hypertension 2009 Oct;54(4):818-24; Abd ElHafeez S. et al. J Nephrol. 2015 Dec;28(6):729-37 ), corroborated by sympathetic activation (Zoccali C. et al. Circulation. 2002 Mar 19;105(11):1354-9), inflammation (Zoccali C. et al. J Am Soc Nephrol. 2002 Feb;13(2):490-6; Mallamaci F. et al. Kidney Int. 2005 Jun;67(6):2330-7.Tripepi G. et al. Clin J Am Soc Nephrol. 2011 Jul;6(7):1714-21;), as well as a metabolic derangement (Zoccali C. et al. Lancet 2001; 358(9299):2113-7; Zoccali C. et al., J Am Soc Nephrol 2002 Jan;13(1):134-41; Mallamaci F. et al. J Am Soc Nephrol. 2004 Feb;15(2):435-41; Testa A. et al. J Bone Miner Res. 2010 Feb;25(2):313-9. These determinants of CV in CKD are no doubt involved in the pathophysiology of arterial stiffness, that is the topic of this paper. Therefore, previous bibliography cannot be omitted in the reference list and should be brought to the fore both in the Introduction and in the Discussion. As to upstroke time, that is a novel predictor of mortality in patients with chronic kidney disease, it looks promising, although whether it provides additional information as compared with pulse wave velocity (PWV) remains so far unclear.

  • Thanks for your great comment. We add the landmark findings in the Introduction and Discussion sections. (page 1, line: 34-37; page 2, line 38-56; page 11)

Specific Comments

  1. Does upstroke time have an additional value as compared with PWV?
  • Thanks for your great comment We compared 3 models, including clinical model, clinical model plus baPWV, and clinical model plus baPWV and UT, in Figure 4. The addition of UT into the clinical model plus baPWV offered an extra benefit in prediction of cardiovascular (Chi-square increase 23.155, P < 0.001) and all-cause mortality (Chi-square increase 28.439, P < 0.001). (page 8, line: 347-356; Figure 4)
  1. PWV was shown to be predicted by pulse pressure in patients with mild to moderate CKD (Stancanelli B. et al. Kidney Blood Press Res. 2007;30(5):283-8). What about the relationship between pulse pressure and upstroke time in your patients with CKD?
  • Thanks for your great comment. baPWV (Unstandardized coefficient β = 0.114, 95% confidence interval [CI] = 0.043 to 0.185) and Pulse pressure (Unstandardized coefficient β = 0.971, 95% CI = 0.762 to 1.180) was significantly correlated with UT in univariate analysis. However, baPWV was not significantly correlated with UT in multivariate analysis when consider different clinical variables. We do not add these findings into the revised manuscript because Reviewer 2 suggests us to remove the Table regarding the determinants of upstroke time in all study patients.
  1. Did you find any difference in upstroke time as related to different classes of drugs (inhibitors of Renin-Angiotensin Aldosterone System, Calcium Antagonists and Beta-Blockers)?
  • Thanks for your great comment. Several classes of antihypertensive medication, including ARB (Unstandardized coefficient β = 8.958, 95% CI = 1.920 to 15.997), CCB (Unstandardized coefficient β = 11.825, 95% CI = 4.813 to 18.837) and diuretic (Unstandardized coefficient β = 8.294, 95% CI = 0.968 to 15.621) were significantly correlated with UT in univariate analysis. But, ACEi and beta blocker were not significantly correlated with UT. We do not add these findings into the revised manuscript because Reviewer 2 suggests us to remove the Table regarding the determinants of upstroke time in all study patients.

Reviewer 2 Report

The authors examined the relationship between a parameter of peripheral artery disease and prognosis among a group of chronic kidney disease (CKD) patients. Overall, the idea is not without interest, but there are major comments, as listed below.

  1. The inclusion and exclusion criteria need to be justified further. First, “random enrollment” (line 53) is unreasonable for a study involving the retrospective analysis of a prospective cohort like this, since there should be defined inclusion criteria on study initiation. In addition, why did the authors exclude those with atrial fibrillation and aortic stenosis? There should be good reasons to exclude these patients, or the authors should provide a comparison between baseline features between excluded patients and enrolled patients to avoid selection bias.
  2. Why did the authors use the higher or the lower values of ABI, BP, baPWV, and upstroke time from bilateral recordings for analysis? It is recommended that the mean values between bilateral recordings be used in analysis to get a better estimate of body-wide peripheral artery status. Please provide results based on this approach.
  3. For statistical analysis, the authors did not check the distribution of their recorded variables and opted directly to use comparison tests suitable for normally distributed ones. This part needs to be corrected with the correct results provided.
  4. The arbitrary use of the median upstroke time as the cutoff value in Table 1 is unjustified. If the distribution of this variable is not linear, this approach will be biased and give incorrect results. The authors should first examine the distribution of their main categorizing variable and choose binary, tertiary, quaternary, or even more division of their main variable followed by analysis.
  5. In survival analyses, the authors further described that they tested several models to find the best cutoff value of upstroke time (line 159 to 160). However, they did not show this vital information as to how they chose that value. What did they mean by “comparing the Chi-square valves of different models” (line 160)? Without clarifying this, the entire results seem of concern. Please provide all results of models you tested a priori, model components and performance, and why chose that value. Comparisons need to be made on rational grounds.
  6. I suggest the authors consider the possibility of collinearity during their regression analyses, since they also acknowledged in the discussion that upstroke time has been reported to be associated with ABI and other peripheral vascular disease parameters. The authors put multiple similar parameters in their regression models without considering the collinearity issue, which can lead to result biases. This must be dealt with seriously.
  7. As the authors confessed in the first paragraph in their discussion, the two major findings revolved around the prognostic utility of upstroke time, the main variable. Then why bother performing analysis focusing on the “determinants” of upstroke time (as in Table 1 and the first several paragraphs in the result section)? This is a very different subject contrary to the prognostic utility of upstroke time. From this viewpoint, the entire manuscript, including methodology and result, needs to be re-organized to better answer their study question. What is the main purpose of this study? Please be aligned. The coherence between the current study aim, method, and result is relatively suboptimal.
  8. The English style of this manuscript is poor with multiple grammatical errors scattering throughout the manuscript. For example, the authors used past tenses consistently in the introduction but actually present tense should be used. There are more in the other parts of the manuscript, including the lack of verbs or altering tenses, etc. Please have this manuscript edit by native English speakers to increase readability.

Author Response

The comments of the reviewer 2 are as follows:

Review 2 comment

  1. The inclusion and exclusion criteria need to be justified further. First, “random enrollment” (line 53) is unreasonable for a study involving the retrospective analysis of a prospective cohort like this, since there should be defined inclusion criteria on study initiation. In addition, why did the authors exclude those with atrial fibrillation and aortic stenosis? There should be good reasons to exclude these patients, or the authors should provide a comparison between baseline features between excluded patients and enrolled patients to avoid selection bias.
  • Thanks for your comment. Study subjects were randomly included from a group of patients arranged for echocardiographic examinations at Kaohsiung Municipal Siaogang Hospital because of suspecting coronary artery disease, heart failure, hypertension, abnormal cardiac physical examination, survey for dyspnea and the pre-operative cardiac function survey. Because the value had a beat-to-beat variation in patients with atrial fibrillation and severe aortic stenosis might have a significant impact on the value of UT, we exclude such patients. (page 2, line: 64-75)
  1. Why did the authors use the higher or the lower values of ABI, BP, baPWV, and upstroke time from bilateral recordings for analysis? It is recommended that the mean values between bilateral recordings be used in analysis to get a better estimate of body-wide peripheral artery status. Please provide results based on this approach.
  • Thanks for your comment. Owing to asymmetric atherosclerotic process in bilateral limbs, lower value of ABI and higher values of BPs, UT, and baPWV might reflect a more real vascular disease status in our patients. Hence, we used higher values of BPs, UT, and baPWV in the present study. (page 3, line: 88-89)
  1. For statistical analysis, the authors did not check the distribution of their recorded variables and opted directly to use comparison tests suitable for normally distributed ones. This part needs to be corrected with the correct results provided.
  • Thanks for your great comment. After normality of continuous variables was determined by Kolmogorov–Smirnov test, appropriate parametric and nonparametric tests were used. (page 4, line: 113-116)
  1. The arbitrary use of the median upstroke time as the cutoff value in Table 1 is unjustified. If the distribution of this variable is not linear, this approach will be biased and give incorrect results. The authors should first examine the distribution of their main categorizing variable and choose binary, tertiary, quaternary, or even more division of their main variable followed by analysis.
  • Thanks for your great comment. Due to non-normality of UT, it was not suitable for cut-off value by using the median value of upstroke time. Therefore, we divided our patients into those with and without all-cause mortality in Table 1. After normality of continuous variables was determined by Kolmogorov–Smirnov test, appropriate parametric and nonparametric tests were used. (page 4, line: 113-116; Table 1)
  1. In survival analyses, the authors further described that they tested several models to find the best cutoff value of upstroke time (line 159 to 160). However, they did not show this vital information as to how they chose that value. What did they mean by “comparing the Chi-square valves of different models” (line 160)? Without clarifying this, the entire results seem of concern. Please provide all results of models you tested a priori, model components and performance, and why chose that value. Comparisons need to be made on rational grounds.
  • Thanks for your great comment. We carefully re-calculated and provided all Chi-square values of different divided models in the table below. We found that the best cut-off value of UT was 183 ms for prediction of all-cause mortality. Figure 3 showed the Kaplan-Meier curves for cardiovascular mortality-free and all-cause mortality-free survival between patients with UT ≥ 183 ms versus UT < 183 ms (both log-rank P < 0001) . (page 8, line: 342-346)

UT models

All-cause mortality

Chi-square values

P value

UT ≧148 vs UT < 148 ms

25.880

<0.0001

UT ≧149 vs UT < 149 ms

27.030

<0.0001

UT ≧150 vs UT < 150 ms

30.444

<0.0001

UT ≧151 vs UT < 151 ms

31.612

<0.0001

UT ≧152 vs UT < 152 ms

34.362

<0.0001

UT ≧153 vs UT < 153 ms

42.512

<0.0001

UT ≧154 vs UT < 154 ms

40.084

<0.0001

UT ≧155 vs UT < 155 ms

40.061

<0.0001

UT ≧156 vs UT < 156 ms

39.769

<0.0001

UT ≧157 vs UT < 157 ms

38.195

<0.0001

UT ≧158 vs UT < 158 ms

35.338

<0.0001

UT ≧159 vs UT < 159 ms

32.162

<0.0001

UT ≧160 vs UT < 160 ms

36.517

<0.0001

UT ≧161 vs UT < 161 ms

38.718

<0.0001

UT ≧162 vs UT < 162 ms

42.457

<0.0001

UT ≧163 vs UT < 163 ms

45.357

<0.0001

UT ≧164 vs UT < 164 ms

48.683

<0.0001

UT ≧165 vs UT < 165 ms

50.202

<0.0001

UT ≧166 vs UT < 166 ms

56.384

<0.0001

UT ≧167 vs UT < 167 ms

55.925

<0.0001

UT ≧168 vs UT < 168 ms

57.238

<0.0001

UT ≧169 vs UT < 169 ms

62.965

<0.0001

UT ≧170 vs UT < 170 ms

64.613

<0.0001

UT ≧171 vs UT < 171 ms

70.761

<0.0001

UT ≧172 vs UT < 172 ms

68.979

<0.0001

UT ≧173 vs UT < 173 ms

66.331

<0.0001

UT ≧174 vs UT < 174 ms

70.812

<0.0001

UT ≧175 vs UT < 175 ms

69.585

<0.0001

UT ≧176 vs UT < 176 ms

74.311

<0.0001

UT ≧177 vs UT < 177 ms

68.181

<0.0001

UT ≧178 vs UT < 178 ms

72.862

<0.0001

UT ≧179 vs UT < 179 ms

69.767

<0.0001

UT ≧180 vs UT < 180 ms

72.292

<0.0001

UT ≧181 vs UT < 181 ms

77.429

<0.0001

UT ≧182 vs UT < 182 ms

78.309

<0.0001

UT ≧183 vs UT < 183 ms

78.689

(the best value)

<0.0001

UT ≧184 vs UT < 184 ms

78.689

<0.0001

UT ≧185 vs UT < 185 ms

69.177

<0.0001

UT ≧186 vs UT < 186 ms

69.177

<0.0001

UT ≧187 vs UT < 187 ms

71.790

<0.0001

UT ≧188 vs UT < 188 ms

70.342

<0.0001

  1. I suggest the authors consider the possibility of collinearity during their regression analyses, since they also acknowledged in the discussion that upstroke time has been reported to be associated with ABI and other peripheral vascular disease parameters. The authors put multiple similar parameters in their regression models without considering the collinearity issue, which can lead to result biases. This must be dealt with seriously.
  • Thanks for your great comment. Due to highly collinear and correlation between ABI and peripheral vascular parameters, ABI was removed from our study.
  1. As the authors confessed in the first paragraph in their discussion, the two major findings revolved around the prognostic utility of upstroke time, the main variable. Then why bother performing analysis focusing on the “determinants” of upstroke time (as in Table 1 and the first several paragraphs in the result section)? This is a very different subject contrary to the prognostic utility of upstroke time. From this viewpoint, the entire manuscript, including methodology and result, needs to be re-organized to better answer their study question. What is the main purpose of this study? Please be aligned. The coherence between the current study aim, method, and result is relatively suboptimal.
  • Thanks for your great comment. Owing to focus on main purpose of mortality prediction, we removed the section of “determinants of UT” in the revised manuscript.
  1. The English style of this manuscript is poor with multiple grammatical errors scattering throughout the manuscript. For example, the authors used past tenses consistently in the introduction but actually present tense should be used. There are more in the other parts of the manuscript, including the lack of verbs or altering tenses, etc. Please have this manuscript edit by native English speakers to increase readability.
  • Thanks for your comment. We had corrected the grammatical errors in the revised manuscript.

Round 2

Reviewer 1 Report

The quality of this paper was significantly improved. No further concern.

Author Response

Dear Editor:

Thank you for the thorough review on our manuscript (reference number: 815544), entitled " Upstroke time as a novel predictor of mortality in patients with chronic kidney disease". All the comments from the editor and reviewers are carefully considered, and the manuscript is revised according to the comments. We appreciate the reviewers’ kind instructions, suggestions, and corrections. For contrast, the corrections or additions are highlighted in red words in Microsoft Word.

Sincerely yours,

Ho-Ming Su, MD, E-mail: [email protected]

Cardiology/Internal Medicine, Kaohsiung Medical University, 100 Shih-Chuan 1st Road, Kaohsiung 807, Taiwan

Fax: (886) (7) 323-4845

The comments of the reviewer 1 are as follows:

Review 1 comment

Comments and Suggestions for Authors

The quality of this paper was significantly improved. No further concern.

  • Thanks for your great comment.

Reviewer 2 Report

The authors did not respond adequately to the previous comments. Several required re-analysis were not done. Please see the following comments.

  1. Please provide the results of all variables comparison between those who were excluded and those who were included in this study, in a new table in the manuscript.
  2. Please provide the K-S test p values for all continuous variables provided in Table 1, so that the readers can assume the distribution of each variable. I did not find such information.
  3. Please provide the collinearity test results of all regression models you provided in this study. Which variable(s) exhibit collinearity? How did you fix them? This should not be done with simple wordings only. 
  4. I suggest the authors consult a biostatistician regarding how the best model can be chose for a defined variable in the survival analysis. The use of chi-square is unclear and confusing. Without this information, I am afraid the analysis results are not credible. 

Author Response

Dear Editor:

Thank you for the thorough review on our manuscript (reference number: 815544), entitled " Upstroke time as a novel predictor of mortality in patients with chronic kidney disease". All the comments from the editor and reviewers are carefully considered, and the manuscript is revised according to the comments. We appreciate the reviewers’ kind instructions, suggestions, and corrections. For contrast, the corrections or additions are highlighted in red words in Microsoft Word.

Sincerely yours,

Ho-Ming Su, MD, E-mail: [email protected]

Cardiology/Internal Medicine, Kaohsiung Medical University, 100 Shih-Chuan 1st Road, Kaohsiung 807, Taiwan

Fax: (886) (7) 323-4845

The comments of the reviewer 2 are as follows:

Review 2 comment

The authors did not respond adequately to the previous comments. Several required re-analysis were not done. Please see the following comments.

  1. Please provide the results of all variables comparison between those who were excluded and those who were included in this study, in a new table in the manuscript.
  • Thanks for your comment. Owing to lack of peripheral vascular examination and laboratory tests in those excluded patients, we could not provide comparison of variables between included and excluded patients. We add it in the limitation section. (page 12 limitation)
  1. Please provide the K-S test p values for all continuous variables provided in Table 1, so that the readers can assume the distribution of each variable. I did not find such information.
  • Thanks for your great comment. The P value for Kolmogorov-Smirnova test were added in revised Table 1. (revised manuscript page 5, table 1)
  1. Please provide the collinearity test results of all regression models you provided in this study. Which variable(s) exhibit collinearity? How did you fix them? This should not be done with simple wordings only. 
  • Thanks for your great comment. Before multivariable analysis, we performed collinearity test. Both in cardiovascular and all-cause mortality analysis, the value of variance inflation factors of systolic blood pressure and pulse pressure were greater than four (Supplementary Table 1). After we withdrew the variable of systolic blood pressure, all values of variance inflation factors in cardiovascular and all-cause mortality analysis were less than four (Supplementary Table 2). Due to high collinearity between systolic blood pressure and pulse pressure, only pulse pressure was selected into multivariable models. The results of multivariable analyses were the same after removing the variable of systolic blood pressure. (revised manuscript page 4, line:118-122; supplementary table 1 and table 2)

Supplementary Table 1. Collinear analysis for all continuous variables in cardiovascular and all-cause mortality models

Continuous variables

Variance inflation factors in cardiovascular mortality model

Variance inflation factors in all-cause mortality model

Age (year)

1.925

1.925

Systolic blood pressure (mmHg)

4.927

4.927

Pulse pressure (mmHg)

5.388

5.388

Heart rate (beats/min)

1.349

1.349

Body mass index (kg/m2)

1.099

1.099

Fasting glucose (mg/dL)

1.098

1.098

Triglyceride (mg/dL)

1.115

1.115

Total cholesterol (mg/dL)

1.128

1.128

Baseline eGFR (mL/min/1.73 m2)

1.127

1.127

Brachial-ankle pulse wave velocity (per 10 cm/s)

2.010

2.010

Upstroke time (ms)

1.546

1.546

Supplementary Table 2. Collinear analysis for all continuous variables after removing systolic blood pressure in cardiovascular and all-cause mortality models

Continuous variables (after removing systolic blood pressure)

Variance inflation factors in cardiovascular mortality model

Variance inflation factors in all-cause mortality model

Age (year)

1.560

1.560

Pulse pressure (mmHg)

1.734

1.734

Heart rate (beats/min)

1.312

1.312

Body mass index (kg/m2)

1.094

1.094

Fasting glucose (mg/dL)

1.090

1.090

Triglyceride (mg/dL)

1.115

1.115

Total cholesterol (mg/dL)

1.127

1.127

Baseline eGFR (mL/min/1.73 m2)

1.127

1.127

Brachial-ankle pulse wave velocity (per 10 cm/s)

1.727

1.727

Upstroke time (ms)

1.469

1.469

  1. I suggest the authors consult a biostatistician regarding how the best model can be chose for a defined variable in the survival analysis. The use of chi-square is unclear and confusing. Without this information, I am afraid the analysis results are not credible. 
  • Thanks for your comment. The best cut-off value of UT for prediction of cardiovascular and all-cause mortality was determined by receiver operating characteristic curve. From receiver operating characteristic curve (Supplementary Figure), the best cut-off value of UT was 180 ms with 46.6% sensitivity and 80.7% specificity for prediction of cardiovascular mortality and 43.2% sensitivity and 88.9% specificity for prediction of all-cause mortality. Figure 3 showed the Kaplan-Meier curves for cardiovascular mortality-free and all-cause mortality-free survival between patients with UT ≥ 180 ms versus UT < 180 ms (both log-rank P < 001). (revised manuscript page 4, line:125-127; page 8, line 337-340; figure 3; supplementary figure)

Supplementary Figure. (in the attached file)

Receiver operating characteristic curves for cardiovascular mortality (A) and all-cause mortality (B) prediction in our study patients.

Round 3

Reviewer 2 Report

The authors have made much improvement to their manuscript. I have some further suggestions: 1. Please put the supplementary figures and tables to the main text. This journal does not place limits on table/figure counts. 2. The English style is not good, especially considering the text style of the newly revised content. I think the authors should ask an English speaking colleague to polish their manuscript.

Author Response

Dear Editor:

Thank you for the thorough review on our manuscript (reference number: 815544), entitled " Upstroke time as a novel predictor of mortality in patients with chronic kidney disease". All the comments from the editor and reviewers are carefully considered, and the manuscript is revised according to the comments. We appreciate the reviewers’ kind instructions, suggestions, and corrections. For contrast, the corrections or additions are highlighted in red words in Microsoft Word.

Sincerely yours,

Ho-Ming Su, MD, E-mail: [email protected]

Cardiology/Internal Medicine, Kaohsiung Medical University, 100 Shih-Chuan 1st Road, Kaohsiung 807, Taiwan

Fax: (886) (7) 323-4845

The comments of the reviewer 2 are as follows:

Review 2 comment

The authors have made much improvement to their manuscript. I have some further suggestions:

  1. Please put the supplementary figures and tables to the main text. This journal does not place limits on table/figure counts.
  • Thanks for your great comment. We put the supplementary figures and tables in the main text of revised manuscript. (line 153-158, page 6; figure 3, table 3, table 4)
  1. The English style is not good, especially considering the text style of the newly revised content. I think the authors should ask an English speaking colleague to polish their manuscript.
  • Thanks for your great comment. The revised manuscript was undergone English language editing by MDPI.
